# Variable effects of transient *Wolbachia* infections on alphaviruses in *Aedes aegypti*

**Brittany L. Dodson**[1], **Sujit Pujhari**[2], **Marco Brustolin**[3], **Hillery C. Metz**[1], **Jason L. Rasgon** [1,4,5,6] *

**1** Department of Entomology, Pennsylvania State University, University Park, Pennsylvania, United States of America, **2** Department of Pharmacology Physiology and Neuroscience, School of Medicine, University of South Carolina, South Carolina, United States of America, **3** Unit of Entomology, Department of Biomedical Sciences, Institute of Tropical Medicine, Antwerp, Belgium, **4** Center for Infectious Disease Dynamics, Pennsylvania State University, University Park, Pennsylvania, United States of America, **5** The Huck Institutes of the Life Sciences, Pennsylvania State University, University Park, Pennsylvania, United States of America, **6** Department of Biochemistry and Molecular Biology, Pennsylvania State University, University Park, Pennsylvania, United States of America

* jlr54@psu.edu

**Data Availability Statement:** All data are available in the figures, tables, and supplementary material

**Funding:** This study was supported by NIH grants R01AI116636 and R01AI150251, USDA Hatch project 4769, a grant with the Pennsylvania

## Abstract

*Wolbachia pipientis* (= *Wolbachia*) has promise as a tool to suppress virus transmission by *Aedes aegypti* mosquitoes. However, *Wolbachia* can have variable effects on mosquito-borne viruses. This variation remains poorly characterized, yet the multimodal effects of *Wolbachia* on diverse pathogens could have important implications for public health. Here, we examine the effects of transient somatic infection with two strains of *Wolbachia* (*w*AlbB and *w*Mel) on the alphaviruses Sindbis virus (SINV), O'nyong-nyong virus (ONNV), and Mayaro virus (MAYV) in *Ae. aegypti*. We found variable effects of *Wolbachia* including enhancement and suppression of viral infections, with some effects depending on *Wolbachia* strain. Both *w*AlbB- and *w*Mel-infected mosquitoes showed enhancement of SINV infection rates one week post-infection, with *w*AlbB-infected mosquitoes also having higher viral titers than controls. Infection rates with ONNV were low across all treatments and no significant effects of *Wolbachia* were observed. The effects of *Wolbachia* on MAYV infections were strikingly strain-specific; *w*Mel strongly blocked MAYV infections and suppressed viral titers, while *w*AlbB had more modest effects. The variable effects of *Wolbachia* on vector competence underscore the importance of further research into how this bacterium impacts the virome of wild mosquitoes including the emergent human pathogens they transmit.

## Author summary

In recent years, wild populations of *Aedes aegypti* mosquitoes in over a dozen countries have been deliberately infected with *Wolbachia pipientis* ("*Wolbachia*"); an intracellular bacterium that, in some circumstances, helps to curb the spread of mosquito-brone pathogens including dengue virus. But how does *Wolbachia* affect the ability of mosquitoes to become infected with and spread the many different viruses they encounter in nature?

Department of Health using Tobacco Settlement Funds, and funds from the Dorothy Foehr Huck and J. Lloyd Huck endowment to JLR. The funders had no role in study design, data collection and analysis, decision to publish, or preparation of the manuscript. All authors had partial salary support from NIH.

**Competing interests:** The authors have declared that no competing interests exist.

Here, we use transient somatic infections in *Aedes aegypti* to characterize the effects of *Wolbachia* on three different alphaviruses that cause illness in humans: Sindbis virus, O'nyong-nyong virus, and Mayaro virus. We find that transient *Wolbachia* infections have variable effects on these different pathogens, ranging from significant suppression of Mayaro virus to significant enhancement of Sindbis virus. Our research has important implications for the design of vector control strategies, and suggests further research is needed to understand how *Wolbachia* shapes the replication and transmission of diverse viruses in mosquitoes.

## Introduction

More than half of the world's population is at risk for vector-borne diseases, with an estimated one billion new infections and one million deaths every year [1]. Vector-borne diseases are an increasing threat to human health due to global travel, insecticide resistance, and climate change [2–5], and novel strategies to combat mosquitoes and the pathogens they transmit are urgently needed. One of the most promising new tools is the bacterium *Wolbachia pipientis* (= *Wolbachia*), which can suppress vector populations [6] and prevent replication of viruses in mosquitoes, an effect called pathogen blocking [7–8].

*Wolbachia* is a genus of intracellular bacteria present in many arthropod species [9–11]. Because it can suppress the transmission of specific mosquito-borne viruses and parasites when transferred to novel mosquito hosts, *Wolbachia* has been the focus of much recent research (e.g., [12–15]). *Wolbachia*-infected mosquitoes have been released into the field in multiple countries to curb the spread of dengue virus (DENV) by *Ae. aegypti* vectors [8,9,16–20]. In some cases, *Wolbachia*-infected animals can replace native populations and retain a pathogen-blocking phenotype for multiple years after release [8,9,21–25]. However, native population replacement with *Wolbachia*-infected mosquitoes is not always successful [16,26–30]. Moreover, the effects of *Wolbachia* on pathogens can be variable and may depend on factors such as the virus–mosquito–*Wolbachia* strain pairing, environmental conditions, population dynamics, and *Wolbachia* density [8,13,31–35]. In several mosquito genera, *Wolbachia* may enhance some pathogens by increasing both infection frequency and infection intensity, including *Plasmodium berghei*, *Plasmodium yoelii*, *Plasmodium gallinaceum*, and West Nile virus (WNV) [35–39]. Our previous work with *Culex tarsalis* demonstrated that a single strain of *Wolbachia* can have different effects on different pathogens. Specifically, the *Wolbachia* strain *w*AlbB enhanced WNV infection frequency but suppressed Rift Valley fever virus titers [39–40]. These findings of enhancement stress the importance of better understanding the multifaceted effects of *Wolbachia* on vectors and pathogens, as *Wolbachia* has the potential to negatively impact mosquito-borne disease control efforts.

To better understand the range of outcomes *Wolbachia* can have on vector competence, we investigated the effects of two *Wolbachia* strains (*w*AlbB and *w*Mel) on alphavirus infections in *Aedes aegypti*. We focused on *Ae. aegypti*, one of the most pernicious vectors of medically relevant pathogens, and to date, the only species used for *Wolbachia* field releases. *Wolbachia* is not naturally found in wild populations of *Aedes aegypti* [41–42]. We studied the alphaviruses Sindbis virus (SINV), O'nyong-nyong virus (ONNV), and Mayaro virus (MAYV). All three viruses are human pathogens and share important characteristics with Chikungunya virus [43–45], an emergent human pathogen spread primarily by *Ae. aegypti* [46]. Infections with these viruses rarely cause mortality, but they do cause significant morbidity (including

fever, rash, and arthralgia) and place a significant burden on public health in affected areas [47–49].

SINV has been isolated from wildlife in Eurasia, Africa, and Oceania [50–51], and there have been periodic cases and epidemics in several areas including Finland, Sweden, Russia, China, Australia, and South Africa [52–57]. Multiple mosquito genera can transmit SINV but *Culex* and *Culiseta* are considered the primary vectors [47,51,58–59]. ONNV is endemic in Africa, where there have been epidemics involving millions of people and where anti-ONNV antibodies are detected at high rates in local human populations [60–64]. ONNV is thought to be transmitted mainly by *Anopheles*, but other mosquito species are also susceptible to infection [65–66]. MAYV is endemic in South and Central America and has caused several small-scale outbreaks of febrile illness with prolonged, disabling arthralgia since it was first identified in 1954 [48]. The virus is common in populations of wild primates and is thought to be spread to humans primarily by *Haemagogus janthinomys* [67], though many mosquito species including *Ae. aegypti* can also become infected and transmit MAYV [49,68–70].

We assessed the ability of transient infections of *w*AlbB and *w*Mel strains of *Wolbachia* to affect infection, dissemination, and transmission of SINV, ONNV, and MAYV in *Ae. aegypti*. We found striking variation in the effects of *Wolbachia* on these viruses, highlighting the need for more research into this bacterium and how it may influence the full diversity of medically relevant arboviruses found in nature.

## Materials and methods

### Mosquitoes, *Wolbachia*, and intrathoracic injections

We used two *Ae. aegypti* colonies. The Rockefeller strain was kindly provided by Dr. George Dimopoulos, Johns Hopkins University, while the Liverpool strain was obtained from BEI resources. Rockefeller mosquitoes were used to test ONNV and SINV, while Liverpool animals were used to test MAYV. All mosquitoes were reared and maintained using standard methods at 27˚C ± 1˚C, 12:12 hr light:dark cycle at 80% relative humidity in 30 × 30 × 30 cm cages (MegaView Science). Larvae were fed Tropical Flakes (Tetramin, Product No. 77101) and adults were provided *ad libitum* access to 10% sucrose. Mosquitoes were fed commercially available expired anonymous human blood (Biological Specialty Corporation) for both virus feeds and colony maintenance.

The *Wolbachia* strains *w*AlbB and *w*Mel (derived from *Ae. albopictus* and *D. melanogaster*, respectively) were purified from infected *Anopheles gambiae* Sua5B cells and resuspended in Schneider's Insect Media (Sigma Aldrich) using published protocols [71]. A cell lysate negative control was prepared by putting *Wolbachia*-negative Sua5B cells through the *Wolbachia* purification process. *Wolbachia* viability and density from cell cultures were assessed by using the LIVE DEAD BacLight Bacterial Viability Kit (Invitrogen) and a hemocytometer.

Two- to five-day-old adult female *Ae. aegypti* were anesthetized with ice and injected in the thorax as previously described [39] with approximately 0.1 µl of *Wolbachia* ($10^{10}$ bacteria/mL) or cell lysate control. Mosquitoes were given access to 10% sucrose *ad libitum* and maintained for up to 22 days post-injection (i.e., up to 27 days of age). *Wolbachia* infection rates in somatically-infected mosquitoes were ~100% and *Wolbachia* titers did not vary across injection groups (ANOVA, F = 1.005, $P$ = 0.39 [S1 Fig]).

### Generation of virus stocks

SINV (p5′dsMRE16ic) and ONNV (p5′dsONNic/foy) plasmids were kindly provided by Dr. Brian Foy (Colorado State University, Ft. Collins, CO) on filter paper [72–73]. We obtained the MAYV strain BeAr505411 from BEI Resources. For SINV and ONNV, infectious virus

stocks were propagated from the plasmid DNA. Specifically, a piece of the filter paper was cut and eluted in 0.1 ml TE buffer for approximately 1 hr. Competent *E. coli* cells (New England Biolabs, #C2987H) were transformed with the eluted plasmid DNA according to the manufacturer's instructions and grown on LB broth selection plates. Colonies were then picked from plates and grown in LB broth overnight at 37°C in a shaking incubator. Plasmid DNA was isolated from the bacterial culture using the EZNA Plasmid Mini Kit (Omega, Cat # D6942-02) according to the manufacturer's instructions. Plasmids were linearized with the AscI enzyme (New England Biolabs, #R0558S) for SINV and NotI enzyme (New England Biolabs, Cat. #R0189S) for ONNV in 0.05-ml reactions, according to the manufacturer's instructions. *In vitro* transcription was performed by using a SP6 polymerase Megascript kit (Ambion, AM1334) for SINV and a T7 polymerase Megascript kit for ONNV (Ambion, AM1330) in 0.02-ml reactions according to the manufacturer's instructions. Cap analog m7G(5′)ppp5′G (Ambion, #AM8048-8052) was used in the transcription reaction, and RNA was purified using a Total RNA kit (Omega, R6834-02; from step 7). Vero or C636 cells were transfected with purified RNA using Transmessenger Transfection Reagent (Qiagen, #301525) according to the manufacturer's instructions (ONNV and SINV), or directly infected with virus particles (MAYV). Cell supernatant was harvested after 24–72 h of incubation and stored in 1 mL aliquots at −70°C.

## Alphavirus infections

Seven (SINV and ONNV) or eight (MAYV) days after *Wolbachia* injections, adult mosquitoes were fed on infectious human blood using a glass membrane feeder jacketed with 37°C water. SINV and ONNV were quantified using plaque assays, while MAYV was quantified using focus-forming assays (see below for specific methods). Mosquitoes were sugar-starved overnight prior to blood feeding. Infectious blood meals were prepared by thawing frozen virus stocks to 37°C and adding it to the blood directly prior to feeding. Final blood meal virus titers were: ONNV– $10^6$ pfu/mL; SINV– $10^5$ pfu/mL; MAYV– $10^7$ ffu/mL. Mosquitoes were allowed to feed for one hour then anesthetized briefly on ice and examined for feeding status, and partially or non-blood fed females discarded. Fully engorged females were randomly divided into two groups and maintained in standard conditions as described above. Infected animals were analyzed at 7 and 14 days post-blood feeding. More specifically, mosquitoes were anesthetized with trimethylamine and legs from each individual were removed and placed separately into 2-ml microcentrifuge tubes containing 1 ml of mosquito diluent (20% heat-inactivated fetal bovine serum [FBS] in Dulbecco's phosphate-buffered saline, 50 μg ml$^{-1}$ penicillin streptomycin, and 2.5 μg ml$^{-1}$ fungizone). Saliva was collected from mosquito bodies by placing the proboscis of each mosquito into a capillary tube containing 1:1 of 50% sucrose:FBS [40]. After 30 minutes, the capillary tube contents were expelled in individual microcentrifuge tubes containing 0.1 ml of mosquito diluent on ice, while bodies were placed in individual microcentrifuge tubes containing 1 ml of mosquito diluent. A single zinc-plated, steel, 4.5 mm bead (Daisy) was placed into the microcentrifuge tubes containing mosquito bodies and legs. SINV and ONNV samples were homogenized in a mixer mill (Retsch) for 30 seconds at 24 cycles per second, then centrifuged for 1 minute at 10,000 rpm. MAYV samples were homogenized at 30 Hz for 2 min in a TissueLyser II (Qiagen) and centrifuged for 30 sec at 11,000 rpm. All samples were stored at −70°C until use.

## Plaque assays

Mosquito samples were tested for SINV or ONNV infectious particles by plaque assay on Vero cells according to previously published protocols [74]. Briefly, 100 μL of each undiluted sample

was inoculated onto Vero cell culture monolayers. After inoculated plates were incubated in a cell culture incubator at 37˚C and 5% $CO_2$ for 1 hr, an agar overlay was added (1:1 1x Dulbecco's modified eagle medium, 10% FBS, 1x penicillin streptomycin, 1x fungizone:1.2% agarose). Plates were incubated at 37˚C for 2 days and then a second overlay (first overlay plus 1.5% final concentration of neutral red) was added. Twenty-four hours after application of the second overlay, samples were scored as positive or negative, and plaques counted. If plaques were too numerous to count, the assays were repeated with 10-fold serial dilutions of the sample.

## Focus forming unit (FFU) assays

Infectious MAYV particles were detected and quantified via FFU assays in Vero cells as previously described [68]. Cells (1x10$^4$/well) were grown in 96-well plates at 37˚C with 5% $CO_2$ in complete media (Dulbecco's modified-essential media [DMEM] with 100 units/mL penicillin/streptomycin and 10% FBS). After one day of incubation, cells were briefly washed with DMEM (without FBS) and incubated for 1 h at 37˚C with 30 uL of the serially diluted ($10^{-1}$ to $10^{-4}$) mosquito lysate or saliva. After 1 h, the sample was removed, and cells were briefly washed with DMEM to remove any unadhered viral particles. Wells were next filled with 100 uL of overlay medium (1% methylcellulose in complete medium), and plates were incubated. After 24 h (body and leg samples) or 48 h (saliva), cells were fixed with 4% paraformaldehyde (Sigma). Fixed cells were blocked and permeabilized for 30 min in blocking solution (3% bovine serum albumin and 0.05% Tween-20 in PBS) then washed with cold PBS. Viral antigens were next labeled with an anti-alphavirus antibody (CHK-48, BEI Resources) diluted 1:500 in blocking solution. Cells were washed with cold PBS four times, then incubated with Alexa-488 tagged secondary antibody (goat anti-mouse IgG, Invitrogen) at a dilution of 1:500. Fluorescent foci were then counted by eye (in a well with a dilution that produced <100 total foci) using an Olympus BX41 microscope with a UPlan FI 4x objective and FITC filter.

## Measurements

Virus infection rate was defined as the proportion of mosquitoes with virus-positive bodies. The dissemination rate was defined as the proportion of infected mosquitoes with virus-positive legs. The transmission rate was calculated as the proportion of animals with disseminated (leg -positive) infections that also had virus-positive saliva, while transmission efficiency was the proportion of total mosquitoes with virus-positive saliva (Fig 1).

## Quantitative real-time PCR of *Wolbachia* density

We extracted DNA from a 250-µl aliquot of each mosquito body homogenate with the EZNA Tissue DNA kit (Omega, cD3396-02), and DNA was used as a template for qPCR with the PerfeCta SYBR FastMix kit (Quanta Biosciences) on a Rotor-Gene Q (Qiagen) or a 7500 PCR system (Applied Biosystems). The qPCRs were performed in 10-µl reactions, and we used the following standardized program for amplification: 95˚C for 5 min; 40 cycles of 95˚C for 10 sec, 60˚C for 15 sec, and 72˚C for 10 sec. DNA was amplified with primers specific to each *Wolbachia* strain (*w*AlbB: Alb-GF; GGT-TTT-GCT-TAT-CAA-GCA-AAA-G and Alb-GR; GCG-CTG-TAA-AGA-ACG-TTG-ATC [75]; *w*Mel: WD_0550F; CAG-GAG-TTG-CTG-TGG-GTA-TAT-TAG-C and WD_0550R; TGC-AGG-TAA-TGC-AGT-AGC-GTA-AA [76]) and was normalized to host gene S7 (AeS7F; GGG-ACA-AAT-CGG-CCA-GGC-TAT-C and AeS7R; TCG-TGG-ACG-CTT-CTG-CTT-GTT-G [77]) by using qGene [39, 78].

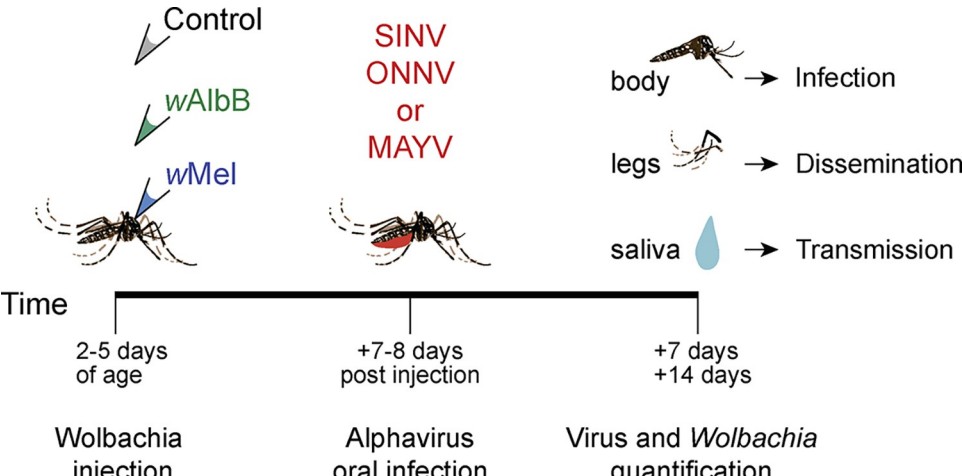

**Fig 1. Schematic of study design and timeline.** Adult *Aedes aegypti* females were somatically infected with *Wolbachia* (*w*AlbB or *w*Mel) or a control solution via injection 2–5 days post-eclosion. Seven or eight days later, injected animals consumed a blood meal spiked with infectious alphavirus (ONNV, SINV, or MAYV). At 7 and 14 days post-blood feeding, viral titers were measured in three tissues. *Wolbachia* infection density was additionally quantified in SINV- and ONNV-exposed animals.

## Statistical analyses

The infection, dissemination, and transmission frequencies for each *Wolbachia* strain and virus combination were compared with controls using pairwise 2x2 Fisher's exact tests. Non-parametric Mann–Whitney U tests were used to compare viral titers when comparing two groups, and the Kruskal–Wallis test with Dunn's correction for multiple comparisons was used to compare experiments with more than two groups. *Wolbachia* titers were analyzed using ANOVA. Statistical tests were performed in GraphPad Prism version 7 for Windows (GraphPad Software, San Diego, CA).

## Results

### *Ae. aegypti* vector competence pilot experiment for alphaviruses SINV and ONNV

Prior to conducting experiments with *Wolbachia*, we first asked whether *Wolbachia*-free *Ae. aegypti* could be infected with ONNV and SINV. We found *Ae. aegypti* was susceptible to infection (17–20% across two replicates, n = 60 total animals) and dissemination (45%, 5 of 11 infected animals) with ONNV, but not transmission (0%). They were susceptible to infection (100% of 60 animals), dissemination (97–100%, at days 7 and 14, respectively), and transmission (23–38%, at days 7 and 14, respectively) with SINV. We did not test MAYV as our previous work found *Ae. aegypti* to be a competent vector of MAYV; At 7 days post infection with BeAr 505411 strain of MAYV, the infection, dissemination and transmission rates were 86.2%, 60% and 6.7% respectively [68], and other work also found *Ae. aegypti* to be susceptible to infection with MAYV [49].

### *Wolbachia* and SINV co-infections

We asked whether somatic *Wolbachia* infections can influence alphavirus infections in *Ae. aegypti*. Infection rates with SINV were moderate across all treatment groups at both time

**Table 1. Effects of *Wolbachia* on alphavirus infection, dissemination, and transmission rates in *Aedes aegypti*.**

| Group | Control (N) | Control Rate | wAlbB (N) | wAlbB Rate | wAlbB P value | wMel (N) | wMel Rate | wMel P value |
|---|---|---|---|---|---|---|---|---|
| SINV Body 7 days | 61 | 0.426 | 123 | **0.642** | **0.007** | 92 | **0.685** | **0.0024** |
| SINV Body 14 days | 34 | 0.559 | 81 | 0.556 | NS | 83 | 0.687 | NS |
| SINV Saliva 7 days | 61 | 0.016 | 123 | 0.049 | NS | 92 | 0.033 | NS |
| SINV Saliva 14 days | 34 | 0.059 | 81 | 0.037 | NS | 83 | 0.108 | NS |
| ONNV Body 7 days | 30 | 0.067 | 90 | 0.111 | NS | 30 | 0.067 | NS |
| ONNV Body 14 days | 26 | 0 | 104 | 0.029 | NS | 38 | 0 | NS |
| ONNV Saliva 7 days | 30 | 0 | 60 | 0.017 | NS | 30 | 0 | NS |
| ONNV Saliva 14 days | 26 | 0 | 66 | 0 | NS | 38 | 0 | NS |
| MAYV Body 7 days | 55 | 0.909 | 40 | 0.8 | NS | 40 | **0.2** | **< 0.00001** |
| MAYV Body 14 days | 57 | 0.842 | 35 | 0.743 | NS | 40 | **0.275** | **< 0.00001** |
| MAYV Legs 7 days | 55 | 0.709 | 40 | **0.35** | **0.0008** | 40 | **0.025** | **< 0.00001** |
| MAYV Legs 14 days | 57 | 0.754 | 35 | **0.457** | **0.0067** | 40 | **0.175** | **< 0.00001** |
| MAYV Saliva 7 days | 52 | 0.115 | 40 | 0.05 | NS | 40 | **0** | **0.0339** |
| MAYV Saliva 14 days | 57 | 0.193 | 35 | **0** | **0.0057** | 40 | **0.025** | **0.0134** |

points (43–69%, Table 1). Both *w*AlbB- and *w*Mel-injected mosquitoes showed significant enhancement of SINV infection rates compared to control mosquitoes at day 7 (Table 1, $P = 0.007$ and $P = 0.002$, respectively) but not at day 14 ($P>0.05$ for both). Neither *Wolbachia* strain affected SINV transmission rates (Table 1). *w*AlbB mosquitoes had significantly greater body titers compared to control mosquitoes at day 7 (Fig 2, $P = 0.004$), with a similar but non-significant trend for wMel mosquitoes (Fig 2). There were no other significant differences in the body or saliva titers between *Wolbachia* and control mosquitoes (Fig 2A and 2B).

### *Wolbachia* and ONNV co-infection

Infection rates with ONNV were low (0–11%) in all treatment groups (Table 1). Neither *Wolbachia* strain had a significant effect on ONNV infection or transmission rates, nor any effects on viral titer (Fig 3, NS for all comparisons).

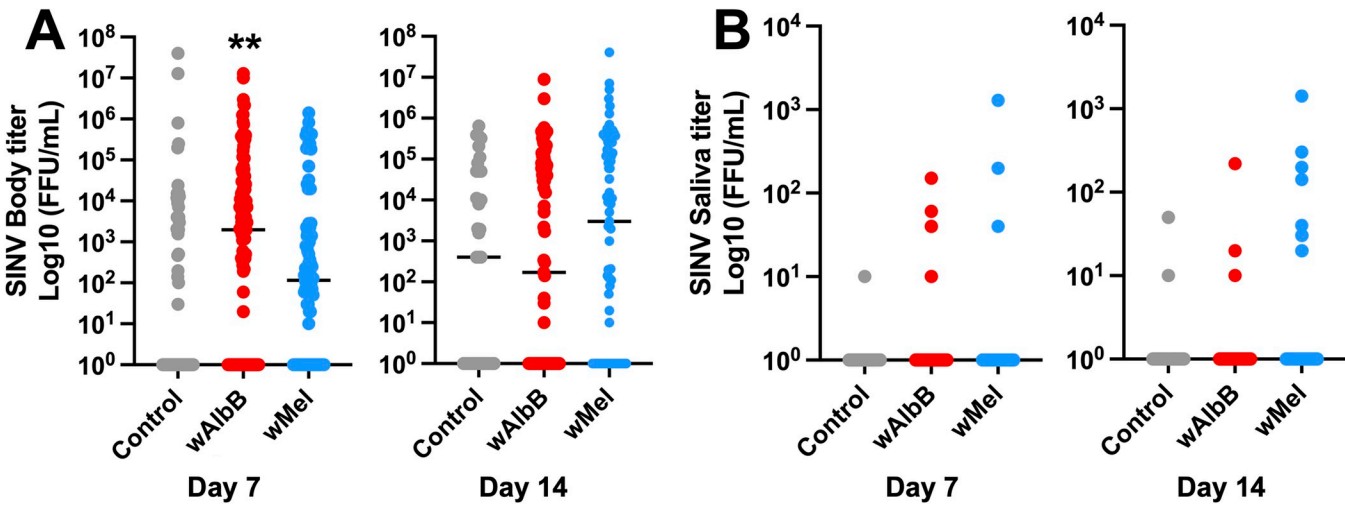

**Fig 2. Effects of *Wolbachia* infection on SINV vector competence in *Aedes aegypti*.** (**A**) SINV body titers at 7 and 14 days post-infected blood meal. (**B**) SINV saliva titers at 7 and 14 days post-infected blood meal. Horizontal lines mark group medians. Groups were compared by Kruskal-Wallis tests with Dunn's correction. ** $P < 0.01$.

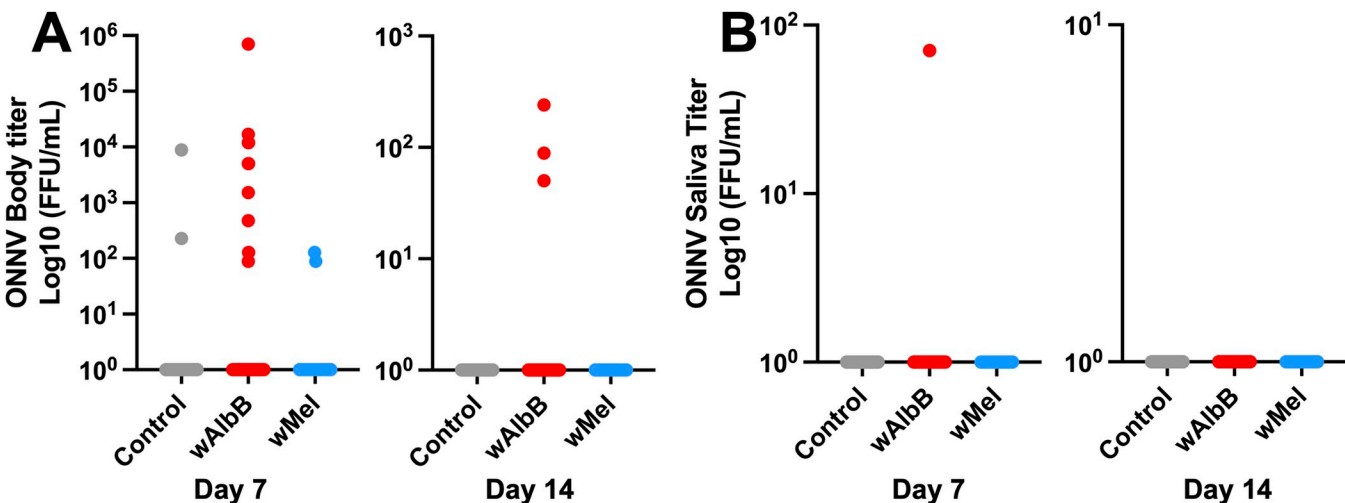

**Fig 3. Effects of *Wolbachia* infection on ONNV vector competence in *Aedes aegypti*.** (**A**) ONNV body titers at 7 and 14 days post-infected blood meal. (**B**) ONNV saliva titers at 7 and 14 days post-infected blood meal. Horizontal lines mark group medians. Groups were compared by Kruskal-Wallis tests with Dunn's correction. There were no significant differences between groups.

### *Wolbachia* and MAYV co-infection

The effects of *Wolbachia* on MAYV infections were *Wolbachia* strain-specific, with greater response to *w*Mel infection compared to *w*AlbB. Control and *w*AlbB-injected mosquitoes were both infected with MAYV at high rates (91% and 80% at day 7, respectively, 84% and 74% at day 14) and did not differ statistically at either time point (Table 1). In contrast, *w*Mel injected mosquitoes were infected with MAYV only rarely (20% and 28% infection rate at days 7 and 14, Table 1, $P<0.00001$ for both time points). MAYV infections were less likely to disseminate in both groups of *Wolbachia* injected mosquitoes (Table 1, $P = 0.0008$ and $P = 0.0067$ for *w*AlbB at days 7 and 14, and $P<0.00001$ for *w*Mel at both timepoints). Transmission was also reduced in most *Wolbachia* injected animals at 7 days post injection for wMel ($P = 0.034$) and for both wAlbB ($P = 0.0057$) and wMel ($P = 0.0134$) at 14 days post injection (Table 1). Both *Wolbachia* strains reduced MAYV infection intensity: *w*Mel had a strong suppressive effect at both time points ($P<0.00001$, Fig 4A) while the effects of *w*AlbB were significant at day 7 ($P = 0.0023$), but while reduced were not significant at day 14 (Fig 4A). Both strains suppressed viral titer in legs—a proxy for dissemination—at both time points (Fig 4B) (*w*AlbB: day 7—$P = 0.008$, day 14—$P = 0.0042$; *w*Mel $P<0.0001$ for both timepoints). *w*AlbB reduced saliva titers only at day 14 ($P = 0.0035$) while *w*Mel reduced saliva titers at both timepoints (day 7: $P = 0.0435$; day 14: $P = 0.0082$) (Fig 4C).

### Discussion

While some mosquito-borne illnesses have declined in recent years (e.g., malaria [79]), *Ae. aegypti*—the primary vector of dengue, yellow fever, chikungunya, and zika viruses—stands out as an increasing threat to global human health [80]. The incidence of dengue, a virus spread primarily by *Ae. aegypti*, has grown 30-fold over the past 50 years and 390 million people may be infected each year [81–82]. *Ae. aegypti* also sparked a new epidemic (Zika virus) by spreading this previously neglected pathogen to new areas of the world [83]. One of the most promising new tools for curbing mosquito-borne disease—and dengue virus in particular—is *Wolbachia*, a bacterium that can block mosquitoes from transmitting pathogens [15]. However, much remains unknown about the bacterium, including its mechanism(s) of action [13,37,84–85]) and how it influences the many diverse viruses *Ae. aegypti* can carry (but see

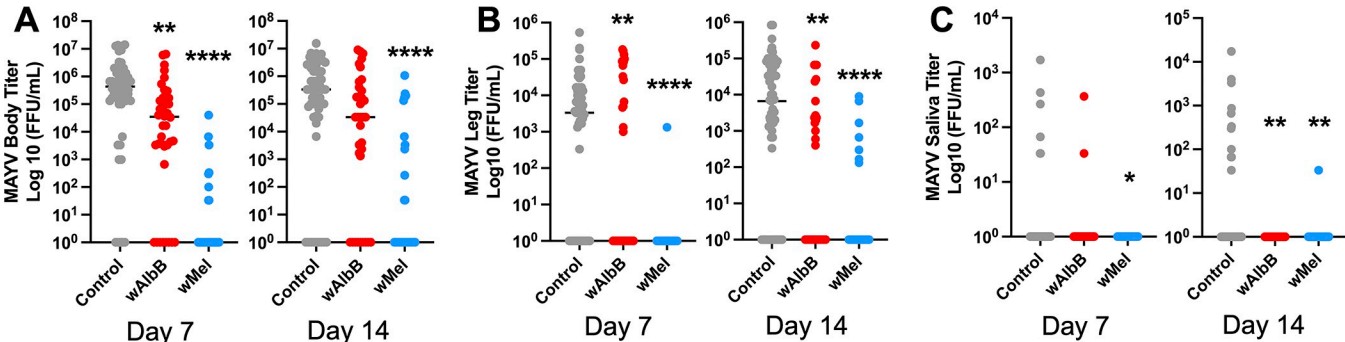

**Fig 4. Effects of *Wolbachia* infection on MAYV vector competence in *Aedes aegypti*.** (**A**) MAYV body titers at 7 and 14 days post-infected blood meal. (**B**) MAYV leg titers at 7 and 14 days post-infected blood meal. (**C**) MAYV saliva titers at 7 and 14 days post-infected blood meal. A-D: Horizontal lines mark group medians. Groups were compared by Kruskal-Wallis tests with Dunn's correction. * $P < 0.05$, ** $P < 0.01$, **** $P < 0.0001$.

[86–89]). Filling these gaps in our understanding will better inform control programs and help us anticipate situations where *Wolbachia* could potentially exacerbate mosquito-borne transmission of some pathogens even while it suppresses others.

Here we report variable effects of *Wolbachia* on different alphaviruses in *Ae. aegypti*. We found that two divergent *Wolbachia* strains enhanced SINV infection rates and titers seven days post infection, though this effect disappeared by day 14. One possibility is that *Wolbachia* decreases the extrinsic incubation period (the time between when a mosquito acquires a virus through a bloodmeal and when it is able to transmit) of this virus, though other mechanisms are possible. In contrast, we did not find significant effects of *Wolbachia* on ONNV infections (though care should be taken in interpreting these data as ONNV infection rates were low in general), and we found *Wolbachia* reduced vector competence for MAYV. This effect varied from strong pathogen blocking (*w*Mel) to smaller effects on dissemination and transmission (*w*AlbB), depending on the *Wolbachia* strain used. Our findings agree with earlier work that reported strong suppression of MAYV by a stable *w*Mel infection in *Ae. aegypti* [49]. In sum, across three different alphaviruses we found three different effects of *Wolbachia*: enhancement, no effect, and strain-dependent pathogen blocking. These disparities highlight that the effects of *Wolbachia* on viruses are extremely variable. With our limited current knowledge, we cannot predict how *Wolbachia* may alter the composition and transmission of *Ae. aegypti*'s large and growing virome [90–91], which includes numerous human pathogens.

We report that *Wolbachia*-mediated effects can be strain-specific. Most notably, the pathogen blocking effect on MAYV [49] depended strongly on the *Wolbachia* strain used: *w*Mel robustly suppressed MAYV infections at both time points while *w*AlbB did not affect infection rate. Our findings comport with previous reports that *Wolbachia* can have strain-specific effects on both pathogen susceptibility phenotypes and immune priming [89, 92–93]. For example, one study in *Ae. aegypti* found the *Wolbachia* strain *w*Mel did not have any effects on yellow fever virus, but the *w*MelPop strain significantly reduced yellow fever virus in mosquito bodies and heads [89]. We do not yet know the mechanism underlying these strain-specific differences. However, we do know that *Wolbachia* strains show substantial genetic variation [6], which may provide one path for uncovering the molecular basis of these differential effects.

Neither *Wolbachia* strain had a significant effect on ONNV vector competence or viral body titers. Though we did see a trend toward possible enhancement of these viral measures, low infection rates affected statistical power. Overall, *Ae. aegypti* in general was a poor vector for ONNV, consistent with reports that *Anopheles* mosquitoes are the main vectors of ONNV [65–66]. However, some studies have suggested that *Ae. aegypti* vector competence for ONNV

may be virus strain-specific, and that this species can be a good vector in some circumstances [66]. Future studies should continue to test *Ae. aegypti* competence for this neglected alphavirus as well as whether *Wolbachia* may enhance ONNV transmission.

Though we do not yet understand why *Wolbachia* has variable effects on diverse viruses including WNV, Rift Valley Fever virus, SINV, ONNV, and MAYV, previous work hints at potential mechanisms. First, viruses from different families may interact within the host and with *Wolbachia* strains in diverse ways, e.g., via distinct immune responses [94]. Another possibility is the nature of the *Wolbachia* infections: this work used transient *Wolbachia* infections rather than stable, maternally inherited *Wolbachia* infections. However, several pieces of evidence suggest a broad similarity between transient and stable infections. Both transient and stable infections can show widespread tissue tropism [21,39,95], and transient and stable *Wolbachia* infections also have similar pathogen-blocking effects on WNV and DENV in *Ae. aegypti* [96]. Transient wMel infections also strongly blocked MAYV infections in the present study, replicating previous findings using stable infections [49]. Thus, the variation we describe may instead arise from previously unexplored biotic or abiotic factors that influence interactions between *Wolbachia* and these pathogens.

Our results illustrate the importance of further research into the effects of *Wolbachia* on arboviruses and the underlying mechanisms of those effects. *Wolbachia* has been deployed widely in the field, yet numerous studies have shown there is substantial variation in the bacterium's effects on vector competence. Factors such as environmental conditions, the *Wolbachia* strain used, the targeted pathogen, the mosquito species, and even rearing conditions appear to influence outcomes (e.g., [97–100]), yet the exact mechanism(s) driving this variation remain unclear. A better understanding of when and how *Wolbachia* influences viral infections in mosquitoes is needed in order to predict the long-range and knock-on effects this bacterium may have on the spread of human pathogens.

There are several limitations to our study. Although transient somatic *Wolbachia* infections have similar effects on both DENV [96] and MAYV [49] in *Ae. aegypti*, it remains to be seen whether (and how) stable *Wolbachia* infections in *Ae. aegypti* affect the alphaviruses studied here. Future work could explore whether *Wolbachia* infection techniques differentially impact pathogens. We also only examined a single viral genotype for each virus and did not compare multiple mosquito genotypes. Finally, we used different mosquito strains for SINV and ONNV experiments compared to MAYV experiments; mosquito genotype can affect *Wolbachia* blocking phenotypes [101–102].

## Supporting information

**S1 Fig.** ***Wolbachia*** **(*w*AlbB and *w*Mel) titers (*Wolbachia* genomes/host genomes) in somatically infected *Aedes aegypti* 7 and 14 days post-injection.** Groups are not statistically different (ANOVA, $P = 0.39$).
(JPG)

**S1 Table. Raw data for this study.** For viral titers, a "1" was added to zero values purely for log-scale plotting.
(XLSX)

## Acknowledgments

We thank Dr. George Dimopoulos of Johns Hopkins University for kindly sharing the Rockefeller strain of *Ae. aegypti* mosquitoes. We thank Dr. Brian Foy of Colorado State University for kindly sharing SINV (p5′dsMRE16ic) and ONNV (p5′dsONNic/foy) plasmids.

## Author Contributions

**Conceptualization:** Brittany L. Dodson, Jason L. Rasgon.

**Data curation:** Brittany L. Dodson.

**Formal analysis:** Brittany L. Dodson, Hillery C. Metz.

**Funding acquisition:** Jason L. Rasgon.

**Investigation:** Brittany L. Dodson.

**Methodology:** Brittany L. Dodson, Sujit Pujhari, Marco Brustolin.

**Project administration:** Jason L. Rasgon.

**Supervision:** Jason L. Rasgon.

**Validation:** Brittany L. Dodson.

**Writing – original draft:** Brittany L. Dodson, Hillery C. Metz, Jason L. Rasgon.

**Writing – review & editing:** Brittany L. Dodson, Hillery C. Metz, Jason L. Rasgon.

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
