## [Decision Letter · Decision Letter 0]

29 Aug 2023

Dear Dr Rasgon,

Thank you very much for submitting your manuscript "Variable effects of Wolbachia on alphavirus infection in Aedes aegypti" for consideration at PLOS Neglected Tropical Diseases. First, let me apologize for a prolonged review process – primarily due to a difficulty in recruiting reviewers, but also due to my leave of absence over the last month. 

Two reviewers provided their thoughtful and detailed assessments of your manuscript, both agreeing on a need for a more explicit statement of the infection model used and a more nuanced discussion on the field implications of your findings. In light of the reviews (below this email), we would like to invite the resubmission of a significantly-revised version that takes into account the reviewers' comments.

We cannot make any decision about publication until we have seen the revised manuscript and your response to the reviewers' comments. Your revised manuscript is also likely to be sent to reviewers for further evaluation.

Sincerely,

Gordana Rasic, Ph.D.

Academic Editor

Álvaro Acosta-Serrano

Section Editor

Dear Dr Rasgon,

Below is the summary of key issues that need to be addressed in your revised manuscript version. 

1. Explicitly emphasizing that the Wolbachia infections in your study were transient (in the title, abstract and introduction).

2. Providing the rationale for testing alphaviruses for which Aedes aegypti has not been implicated as an important vector.

3. Providing the sample sizes for each experimental group in the text and figures; discussing the power to detect the Wolbachia effects given the sample sizes.

Editor’s comment:

One experimental approach that comes to mind, which would be, in my opinion, more rigorous for testing the Wolbachia effects is the use of mosquito strains derived from the recently caught field mosquitoes infected with wMel or wAlbB and comparing them to their cured and transiently-infected counterparts. Could you discuss why such (or similar) experimental design was not employed?

Comments from the reviewers:

**Key Review Criteria Required for Acceptance?**

**Methods**

-Are the objectives of the study clearly articulated with a clear testable hypothesis stated?

-Is the study design appropriate to address the stated objectives?

-Is the population clearly described and appropriate for the hypothesis being tested?

-Is the sample size sufficient to ensure adequate power to address the hypothesis being tested?

-Were correct statistical analysis used to support conclusions?

-Are there concerns about ethical or regulatory requirements being met?

Reviewer #1: Comments regarding the methods according to some of the questions above.

"Rockefeller

mosquitoes were used to test ONNV and SINV, while Liverpool animals were used to test

MAYV". Why not using the same aegypti strain? Although lab strain, any impact on the results? Limitation of the study as these are lab strains and their vector competence would not be the same if field strains were tested…

"Generation of virus stock": any interest or preference to use SINV and OONV derived from plasmid construction compared to MAYV obtained from virus stock. Impact on the results? Limitation of the study.

Reviewer #2: The authors reported variable effects of Wolbachia depending on the strain and the alphavirus tested. The results ranged from wMel strongly blocking Mayaro virus infections and suppressing viral titers to no significant effects observed for O’nyong-nyong virus. Interestingly, there was also an enhancement of Sindbis virus infection in wAlbB- and wMel-infected mosquitoes. Overall, the experiments were well-controlled and appropriate for addressing the objectives of the study. Adequate statistical analyses were performed, although it would be helpful to include the sample size in the figures or mention it in the figure legends.

**Results**

-Does the analysis presented match the analysis plan?

-Are the results clearly and completely presented?

-Are the figures (Tables, Images) of sufficient quality for clarity?

Reviewer #1: Yes

Specific comments

"Ae. aegypti vector competence for alphaviruses SINV and ONNV". Please add a figure in sup file of this control experiment.

The results presented in the figures 2 and 3: VC results of the control strain (control experiment) is not in the same range of the controls presented in the figures except for MAYV. Reproducibility of the results. 

A bias in the experimental set-up is to have used two different aegypti strains. It would have been interesting to test at least once a virus in both aegypti strains, to exclude confounding impact

Reviewer #2: The analysis presented in the manuscript is in accordance with the planned analysis, and the researchers followed their analysis plan to ensure consistency and reliability in their findings. The results are thoroughly and effectively presented, leaving no ambiguity in the information provided. The authors have taken great care in clearly conveying their findings, enabling readers to grasp the outcomes of the study with ease. The figures exhibit a high standard of quality, facilitating clarity in the presentation of data. The visual elements are well-designed, enhancing the understanding of the results. However, it is worth noting that the sample size is missing in all figures or figure legends, which would be helpful for better contextualizing the findings.

**Conclusions**

-Are the conclusions supported by the data presented?

-Are the limitations of analysis clearly described?

-Do the authors discuss how these data can be helpful to advance our understanding of the topic under study?

-Is public health relevance addressed?

Reviewer #1: Specific comments on conclusions.

"Another

possibility is the nature of the Wolbachia infections: this work used transient Wolbachia

infections rather than stable, maternally inherited Wolbachia infections". Fig S1 showed an overall Wolbachia density. However full bodies have been analyzed. It would have been interesting to test legs or heads or… to be sure that injected Wolbachia in the thorax indeed disseminated in the whole mosquito including abdomen/midgut. Indeed with oral blood feeding, viruses has to pass through the midgut barriers and competition between Wolbachia and virus at this step is of prime importance.

"We also only examined a single viral genotype

for each virus and did not compare multiple mosquito genotypes". And more Ae/MAYV results are different from SINV/ONNV/Ae: the last two experiments were conducted with both virus derived from plasmids and rockfeller strain. Limitations of the work. 

"Future work could explore whether Wolbachia infection

techniques differentially impact pathogens. We also only examined a single viral genotype

for each virus and did not compare multiple mosquito genotypes". I do agree.

Reviewer #2: The conclusions drawn in the manuscript are well supported by the data presented. The researchers have conducted a thorough analysis and interpretation of the data, resulting in meaningful conclusions that align with the findings of the study. The authors have effectively described the limitations of the analysis, transparently acknowledging potential constraints and challenges encountered during the study. One notable limitation is the transient nature of the Wolbachia infection, which is crucial to understanding the scope and implications of the research. The authors discuss how the data presented can contribute to advancing our understanding of Wolbachia-mediated viral protection in Aedes aegypti. Their insights into the intricate dynamics between Wolbachia strains, alphaviruses, and Aedes aegypti mosquitoes offer valuable information for future research in the field of controlling vector-borne diseases.

However, it is important to note that the comparison and extrapolation of effects from transient Wolbachia infection to the permanent infection used in Aedes aegypti field strategies may have been oversimplified. Cautious approach is warranted when applying the study's findings in practical applications. This concern is supported by the fact that the importance of Aedes aegypti as a vector varies among the different alphaviruses tested. Hence, considering the context-specific nature of these interactions is essential for a comprehensive understanding.

**Editorial and Data Presentation Modifications?**

Reviewer #1: (No Response)

Reviewer #2: I have several suggestions and comments that I believe the authors should address in their manuscript.

Firstly, it is crucial for the authors to explicitly emphasize that the Wolbachia infection in their study was transient. This distinction is significant because although some studies have reported similar effects between transient Wolbachia infection and permanent infection, it is essential to avoid generalizing the findings to permanent infection scenarios. I recommend that the authors clearly state the transient nature of the Wolbachia infection the manuscript title and mentioning it in abstract to provide readers with a comprehensive understanding of the study design.

By addressing these points, the authors can provide a more accurate and nuanced perspective on the implications of their research, ensuring that readers fully understand the context and limitations of their findings.

Minor comments:

- Include the word transient in the title

- Include the word transient in the abstract

- Include smaple size in the figures or in the figure legends

- Given that the authors observed a strong blocking effect of wMel on Mayaro virus (MAYV) infections, and MAYV is the only alphavirus known to potentially be transmitted by Aedes aegypti in the field, it is crucial for the authors to discuss the implications of their findings more cautiously. Unlike Sindbis virus (SINV) and O'nyong-nyong virus (ONNV), where Aedes aegypti does not play a significant role in transmission to humans, the blocking effect observed in the case of MAYV has direct relevance to arbovirus transmission. The authors should discuss the limitations and caveats associated with extrapolating these findings to other alphaviruses and real-world scenarios.

**Summary and General Comments**

Reviewer #1: This work describes the vector competence for SINV, ONNV and MAYV for Ae aegypti transiently infected or not with wMel and wAlbB. Although interesting this work raised me a major remark: Why did the authors wanted to test these viruses on Ae. aegypti. Ae. Aegypti is not the major vector of these viruses. Studies have been shown that Ae. Aegypti can transmit SINV and ONNV, but it is very few and rely on specific Virus/Vector interactions. Except, for MAYV, Ae. Aegypti has been described as a potential vector along with lots of other one. Thus I am interrogating myself about the public heath interest of such experiments. Could the authors clearly justify their purposes and hypotheses.

Specific comments:

Introduction:

“In some cases, Wolbachia-infected animals….”: Why in some cases, if not in all cases please precise or adjust your sentence. Animals should be replaced by mosquitoes.

“However, native population replacement with Wolbachia-infected mosquitoes is not always successful

[16, 26-30].” These refs do not refer to unsuccessful replacement.

"Because previous studies have focused primarily on flaviviruses while neglecting

alphaviruses, we studied the alphaviruses Sindbis virus (SINV), O’nyong-nyong virus

(ONNV), and Mayaro virus (MAYV"). Indeed, but for none of these viruses, Ae. aegypti is an important vector.

"Multiple mosquito genera can transmit SINV butCulex and Culiseta are considered the primary vectors [47, 51, 58-59]." What about aedes? Same for ONNV?

Saying that:" In

sum, these alphaviruses are a burden on global human health yet remain poorly understood,

including if and how they are affected by Wolbachia". It would have been more interesting to test the effect of Wolbachia (is possible) in their major vectors.

Reviewer #2: In this manuscript, Dodson et al. investigated the effects of a transient infection of two Wolbachia strains (wAlbB and wMel) on alphavirus protection in Aedes aegypti mosquitoes. Overall, the experiments were well-controlled and appropriate for addressing the objectives of the study. Adequate statistical analyses were performed, although it would be helpful to include the sample size in the figures or mention it in the figure legends.

PLOS authors have the option to publish the peer review history of their article (what does this mean?). If published, this will include your full peer review and any attached files.

Reviewer #1: No

Reviewer #2: Yes: Alvaro Gil Araujo Ferreira
---

## [Decision Letter · Decision Letter 1]

5 Feb 2024

Dear Dr. Rasgon,

Thank you very much for submitting your manuscript "Variable effects of transient Wolbachia infections on alphaviruses in Aedes aegypti" for consideration at PLOS Neglected Tropical Diseases. As with all papers reviewed by the journal, your manuscript was reviewed by members of the editorial board and by two independent reviewers. 

The reviewers have minor comments and requests for clearer definitions and some figure edits. We are happy to accept the manuscript and proceed with the publication process pending these minor edits.

Sincerely,

Gordana Rasic, Ph.D.

Academic Editor

Álvaro Acosta-Serrano

Section Editor

Dear Dr. Rasgon,

The reviewers have minor comments and requests for clearer definitions and some figure edits. We are happy to accept the manuscript and proceed with the publication process pending these minor edits.

Reviewer's Responses to Questions

**Key Review Criteria Required for Acceptance?**

**Methods**

-Are the objectives of the study clearly articulated with a clear testable hypothesis stated?

-Is the study design appropriate to address the stated objectives?

-Is the population clearly described and appropriate for the hypothesis being tested?

-Is the sample size sufficient to ensure adequate power to address the hypothesis being tested?

-Were correct statistical analysis used to support conclusions?

-Are there concerns about ethical or regulatory requirements being met?

Reviewer #3: (No Response)

Reviewer #4: - Are the objectives of the study clearly articulated with a clear testable hypothesis stated?

YES, the study meets these criteria

-Is the study design appropriate to address the stated objectives?

YES, but please provide some information on the Wolbachia purification protocol, or at least refer to a published paper. 

-Is the population clearly described and appropriate for the hypothesis being tested?

YES

-Is the sample size sufficient to ensure adequate power to address the hypothesis being tested?

YES

-Were correct statistical analysis used to support conclusions?

YES

-Are there concerns about ethical or regulatory requirements being met?

NO

**Results**

-Does the analysis presented match the analysis plan?

-Are the results clearly and completely presented?

-Are the figures (Tables, Images) of sufficient quality for clarity?

Reviewer #3: (No Response)

Reviewer #4: -Does the analysis presented match the analysis plan?

YES

-Are the results clearly and completely presented?

Generally yes, but it would have been good to have some information on the tissues where Wolbachia ended up following infection. 

-Are the figures (Tables, Images) of sufficient quality for clarity?

YES

**Conclusions**

-Are the conclusions supported by the data presented?

-Are the limitations of analysis clearly described?

-Do the authors discuss how these data can be helpful to advance our understanding of the topic under study?

-Is public health relevance addressed?

Reviewer #3: (No Response)

Reviewer #4: -Are the conclusions supported by the data presented?

YES

-Are the limitations of analysis clearly described?

YES, in the revised document. The authors should make it clear (page 6, 2nd paragraph) that they are testing transient infection. 

-Do the authors discuss how these data can be helpful to advance our understanding of the topic under study?

YES generally

-Is public health relevance addressed?

The public health relevance is limited as transient infections will not be deployed in the field. However, the observations of this study are worthy of publication in the general context of Wolbachia biology.

**Editorial and Data Presentation Modifications?**

Reviewer #3: (No Response)

Reviewer #4: (No Response)

**Summary and General Comments**

Reviewer #3: Dodson et al investigated the effect of two Wolbachia strains in Aedes aegypti on three different alphavirus infections. While it is noted that Ae. aegypti are not the major vector of these viruses in the field, the study highlights the importance of examining the effect of various wolbachia strains in various mosquito species against different viral families, due to the potential for enhanced viral infection in mosquito species by viral families that may not be usually associated with those mosquito species. The experimental design is sound for the conclusions drawn from the study, and the manuscript is overall well written. I was not involved in the previous round of review, but responses to the comments appear sound, although lack of a tracked manuscript makes it difficult to determine what changes were made in response to some comments. I suggest the following changes to the manuscript:

1) The definitions of infection rate, dissemination rate, transmission rate, and transmission efficiency are buried in the plaque assay methods section. Figure legends, results text, figure 1, and perhaps graph axis labels should all be changed to more clearly define what these are actually measuring to make it more readily accessible to the reader. E.g. “transmission rate”. Transmission to what? Dissemination rate – dissemination to what? Some may automatically assume dissemination to salivary gland, for example. Percentage of virus positive bodies, legs, and salivary glands would be an easier way to describe the data. Similarly, “infection intensity” should be changed to “viral titers in bodies and saliva”.

2) Figures can easily be combined, to have 1 figure for each virus, especially since the ‘rates’ are calculated from the same data presented in the following figure. I.e. merge figure 2 and 3 together, 4 and 5 together, and 6 and 7 together.

3) Why were the viral titers in blood meals different between ONNV, SINV and MAYV, and could this affect the effect of Wolbachia between viral strains?

4) Check throughout that the use of “strain-specific” is clearly defined to avoid confusion between wolbachia strain versus virus strain. E.g. see line 2 in “Wolbachia and MAYV co-infection” results section.

5) Fig 7C day 14 control versus wAlbB – presumably significant?

6) Define “extrinsic incubation period of this virus” in the discussion.

Reviewer #4: I did not review the original manuscript, but a revised version. Looking at the revisions and author replies, it is clear that they have addressed the previous concerns. I do have a few additional comments: 

Line numbers would have been useful for this manuscript

Page 5, paragraph 2: it’s incorrect to say that less attention has been paid to alphaviruses in this context; such literature has been there from the early days of this field and more recently (e.g. Moreira et al. 2009 chikungunya; van den Hurk et al. 2012 chikungunya; Pereira et al. 2018 Mayaro; Battacharya et al. 2017 sindbis in Drosophila; Ekwudu et al. 2020 sindbis and some other alphas).

Page 6, 2nd line: a lot is in fact known about sindbis (it’s a workhorse of alphavirus biology), but not about ONNV or MAYV; I suggest rephrasing here. 

My main concern is that transient infections don't really represent the interaction between Wolbachia and RNA viruses, as the infection itself is likely to lead a very large immune response and general antimicrobial state that could obscure the true extent of virus inhibition that may occur with a permanently infected line. 

The lead author raises the point that not all researchers have access to lines with established Wolbachia infections (especially wMel due to legal issues), and this situation is unfortunate. Perhaps the lead authors could explore other avenues in the future, for example wAlbB from Professor Xi.

PLOS authors have the option to publish the peer review history of their article (what does this mean?). If published, this will include your full peer review and any attached files.

Reviewer #3: No

Reviewer #4: No

Figure Files:

Data Requirements:

Reproducibility:

References

---

## [Editor Report · Decision Letter 2]

15 Oct 2024

Dear Dr Rasgon,

We are pleased to inform you that your manuscript 'Variable effects of transient Wolbachia infections on alphaviruses in Aedes aegypti' has been provisionally accepted for publication in PLOS Neglected Tropical Diseases.

Best regards,

Gordana Rasic, Ph.D.

Academic Editor

Álvaro Acosta-Serrano

Section Editor

---

## [Editor Report · Acceptance letter]

28 Oct 2024

Dear Dr. Rasgon,

We are delighted to inform you that your manuscript, "Variable effects of transient Wolbachia infections on alphaviruses in Aedes aegypti," has been formally accepted for publication in PLOS Neglected Tropical Diseases.

Best regards,

Shaden Kamhawi

co-Editor-in-Chief

Paul Brindley

co-Editor-in-Chief
